# Novel, Blended Polymeric Microspheres for the Controlled Release of Methotrexate: Characterization and In Vivo Antifibrotic Studies

**DOI:** 10.3390/bioengineering10030298

**Published:** 2023-02-27

**Authors:** Layla Nabai, Aziz Ghahary, John Jackson

**Affiliations:** 1BC Professional Fire Fighters’ Burn & Wound Healing Research Lab, ICORD, The Blusson Spinal Cord Centre, 818 West 10th Ave, Vancouver, BC V5Z 1M9, Canada; 2Faculty of Pharmaceutical Sciences, The University of British Columbia, 2045 Westbrook Mall, Vancouver, BC V6T 1Z3, Canada

**Keywords:** methotrexate, PLGA, microsphere, fibrosis

## Abstract

Low dose methotrexate (MTX) is known to effectively decrease type I collagen production in dermal fibroblasts, while increasing the matrix metalloproteinase-1 (MMP-1) production in vitro. For in vivo use as an antifibrotic agent on wounds, a linear and extended controlled release formulation of MTX is required. The objective of this study was to optimize the fabrication of MTX-loaded polymeric microspheres with such properties, and to test the efficacy for the prevention of fibrosis in vivo. Poly lactic-co-glycolic acid (PLGA), Poly (L-lactic acid) (PLLA) and the diblock copolymer, methoxypolyethylene glycol-block-poly (D, L-lactide) (MePEG-b-PDLLA), were used to fabricate microspheres, which were then characterized in terms of size, drug encapsulation efficiency, and in vitro release profiles. The optimized formulation (PLGA with diblock copolymer) showed high drug encapsulation efficiency (>80%), low burst release (~10%) and a gradual release of MTX. The amphipathic diblock copolymer is known to render the microsphere surface more biocompatible. In vivo, these microspheres were effective in reducing fibrotic tissue which was confirmed by quantitative measurement of type I collagen and α-smooth muscle actin expression, demonstrating that MTX can be efficiently encapsulated in PLGA microspheres to provide a delayed, gradual release in wound beds to reduce fibrosis in vivo.

## 1. Introduction

Methotrexate (MTX) is a folate antagonist that has been used in the treatment of a wide range of diseases since its first development in 1947 [1]. While high doses of MTX, with antiproliferative effects on cancerous cells, is used in the treatment of malignancies, low doses have been approved by FDA and Health Canada as a disease-modifying agent for the treatment of rheumatoid arthritis (RA) and psoriasis [2,3]. Low dose MTX has also been used in other immune-mediated inflammatory diseases such as Crohn’s disease [4], as a steroid-sparing adjuvant in systemic lupus erythematosus (SLE) [5] and dermatomyositis [6], and for the successful prevention and treatment of fibroproliferative disorders like keloids [7,8,9]. Although the exact mechanism involved in the anti-inflammatory effect of low dose MTX is not elucidated, an increase in the extracellular level of adenosine and inhibition of polyamines have been suggested as possible contributing factors [3]. To reduce the systemic toxic effects of oral or parenteral MTX, such as bone marrow suppression, nausea, and stomatitis, MTX containing formulations for topical use in psoriasis or intra-articular injection in RA have been studied [10,11,12]. However, the barrier function of the epidermis limits trans-epidermal penetration of the MTX from hydrophilic gels [13], and locally injected MTX is rapidly cleared from the intra-articular space [12]. In the last decade, a variety of drug delivery systems such as nanostructure lipid carriers of MTX or polymeric microparticles have been studied for controlled delivery of an effective dose to the targeted tissues [14,15]. The majority of those drug delivery systems were designed for immediate and extended release of the MTX to overcome the frequency of the administration of the drug [16].

Fibrotic scar formation following surgical procedures or medical device implantation is a clinically challenging problem with no satisfactory preventive or therapeutic modalities. The imbalance between two key components of the extracellular matrix (ECM), type I collagen and matrix metalloproteinase-1 (MMP-1), and a defective remodeling phase of the wound healing are the major contributing factors to fibrosis. Fibroblasts and α- smooth muscle actin (α-SMA) expressing myofibroblasts are the main cells contributing to ECM production and tissue contracture in wound healing [17]. A considerable body of evidence has shown that chronic inflammation accompanied by persistent production of growth factors and fibrogenic cytokines stimulates the phenotypic transformation of various cells to myofibroblasts [18,19,20]. As such, an anti-metabolite, anti-inflammatory drug such as MTX might allow for reduced cellularity and collagen deposition in a wound bed during the later stages of wound repair; that might be monitored in vivo by investigations of drug effects on total cellularity, collagen deposition, α-SMA and MMP-1 expression. Our group has previously shown that low dose MTX modulates the expression of type I collagen and MMP-1 in dermal fibroblasts [21]. However, for the successful prevention of fibrosis without any delay in wound healing the timing of drug delivery and dosage of MTX in wound bed is crucial.

The polymers PLLA and PLGA are commonly used for drug delivery because of their biodegradability and biocompatibility properties. Several properties of the polymer and the nature of the encapsulated substance together with the size, shape and porosity of the polymeric drug delivery system influence the release of the encapsulated drug [22]. Drug release from hydrophobic PLGA and PLLA polymers has been shown to be very slow [23]. To improve the drug delivery properties alternative types of block copolymers of polyesters (PLGA) and poly ethylene glycol (PEG), as diblock (PLGA-PEG) or triblock (PLGA-PEG-PLGA), have been developed [24]. An alternative strategy has been to blend similar amphipathic diblock copolymers into PLGA drug-loaded formulations to modify drug release characteristics [23].

In this study we developed and characterized MTX-loaded microspheres using PLLA, PLGA, and MePEG-PDLLA diblock polymers. Following determination of the PLGA 75% + MePEG diblock 20% as the optimal formulation, the anti-fibrotic effect of MTX-loaded microspheres was studied.

## 2. Materials and Methods

### 2.1. Materials

MTX hydrate, Weigert’s Iron Hematoxylin set (HT1079), Trichrome stain (Masson) (HT15-1KT), and trans-4-hydroxy-L-proline, were purchased from Sigma (St. Louis, MO, USA). Poly (vinyl alcohol) (PVA) in the form of 98% hydrolyzed, MW 25,000 Da (Polysciences, Warrington, UK) and PLGA (85/15, IV = 0.61 dL/g) (Birmingham Polymers, Birmingham, AL, USA) were used as supplied. MePEG-b-PDLLA (60:40 *w*/*w*) (MW 3500 Da, measured by GPC) was from in-house stores [23]. Poly (L-lactic acid) (PLLA) (IV: 0.1–0.2 dL/g, MW 2000 Da) was obtained from Polysciences, Warrington, UK. Dichloromethane (DCM), methanol (HPLC grade), sodium chloride, acetonitrile, sodium phosphate monobasic and Triethylamine (TEA) were from Thermo Fisher (Toronto, Canada). Proteinase K, DAPI, and Phenylisothiocyanate (PITC, Edman’s Reagent), were bought from Thermo Scientific (Rockford, IL, USA). Note: the molecular weight of some polymers is described by manufacturers in terms of inherent viscosity (IV).

Antibodies were purchased as follows: Rabbit polyclonal α-SMA antibody (ab 5694) was obtained from Abcam Inc. (Cambridge, MA, USA) and the secondary biotinylated goat anti-rabbit antibody and NovaRED (Vector Labs Burlington, ON, CA) used for immunohistochemical staining. For Western blot methods, mouse-anti-human procollagen type1α 1 aminopropeptide antibody (Developmental Studies Hybridoma Bank, Iowa city, CA, USA), rabbit-mono-anti-human MMP-1(Epitomics, Burlingame, CA, USA) and mouse mono-anti-human β-actin antibody (Sigma) were used as primary antibodies. Goat-anti-rabbit IgG and anti-mouse IgG were horseradish peroxidase conjugated secondary type (Bio-Rad. Mississauga, ON, Canada).

RNA concentration was measured by UV spectrometry. First Strand cDNA Synthesis kit was from Superscript II Invitrogen and SYBR^®^ Green kit was bought from Applied Biosystems, Warrington, UK. Quantitative PCR (qPCR) was performed using Applied Biosystems^®^ 7500 Fast Real-Time PCR System.

The forward and reverse primers for rat MMP-13 were (5′-TTGTTGCTGCCCATGAGCTT-3′) and (5′-ACTTTGTCGCCAATTCCAGG-3′), respectively. The forward and reverse primers for rat Col-1α1 were (5′-CAAGAATGGCGACCGTGGT-3′) and (5′-GGTGTGACTCGTGGAGCCA-3′), respectively. The forward and reverse primers for rat α-SMA were (5′-ACTGGGACGACATGGAAAAG-3′) and (5′-CATCTCCAGAGTCCAGCAGA-3′), respectively. The forward and reverse primers for Rat β-actin were (5′-TATCGGCAATGAGCGGTTCC-3′) and (5′-GTGTTGGCATAGAGGTCTTTACG-3′), respectively (reference gene).

Microspheres were fabricated using a Caframo overhead stirrer (Wiarton, ON, Canada), and morphology of the fabricated microspheres was examined using Polarized light microscopy and a Hitachi S-3000 N scanning electron microscope (SEM).

UV–Vis spectrophotometry was used to quantitate MTX at 300 nm. HPLC quantitation of hydroxyproline was performed using a Waters ACQUITY system (UV detection at 260 nm), Novapak^®^ C18 column (Waters, Mississauga, ON, Canada) with a mobile phase of 140 mM sodium acetate, 0.05% TEA and 6% Acetonitrile, pH 6.4, as eluent at 1 mL/min.

Pre-cut, 1 cm in diameter, polyvinyl alcohol (PVA) sponges were from Medtronic, Inc. (Jacksonville, FL, USA) and used for in vivo experiments.

### 2.2. Fabrication of MTX-Loaded Microspheres

The microspheres were prepared using emulsion/solvent evaporation technique as previously described [25]. Briefly, polymer (e.g., PLLA (2K)) and MTX at a ratio of 95:5 were dissolved in dichloromethane (DCM) at a final concentration of 20% *w*/*v*. Two mL of this solution was then slowly dispersed into 100 mL of a 1% PVA solution while stirring at 600 rpm using an overhead stirrer (BDC 2002 Caframo, Wiarton, Ont., Canada). After 10 min the stirring speed was reduced to 450 rpm and continued for two hours. The hardened microspheres sedimented and were washed 3 times and air dried overnight. The same procedure was used to make MTX-loaded microspheres with different formulations (PLLA: diblock: MTX 75:20:5, PLLA: PLGA: MTX 75:20:5, PLGA: MePEG-PDLLA diblock: MTX 75:20:5).

The same procedure was used for fabrication of polymer-only (empty) microspheres.

### 2.3. Characterization of the Fabricated Microspheres

Polymer-only and MTX-loaded PLLA, PLLA + MePEG diblock, PLLA + PLGA, and PLGA + MePEG diblock microspheres were coated with gold under vacuum and SEM was used to examine the microspheres. SEM and polarized light microscope images for MTX-loaded microspheres were taken. A particle size analyzer (Malvern) was used to examine the particle size of the microspheres as previously described [14]. Briefly, the microspheres were homogenously dispersed in distilled water with two drops of 2% polysorbate 80 (Tween 80) in water. n = 4 for surface weighted mean.

### 2.4. Quantification of MTX Using Spectrophotometer

MTX calibration solutions were prepared in water. First, 3 mg of MTX was dissolved in 1 mL of dimethyl sulfoxide (DMSO). Then, serially diluted solutions of 3 mg MTX in DMSO were made in the range 48, 24, 12, 6, 3, 1.5, 0.75, 0.37 μg/mL of MTX in water.

### 2.5. Evaluation of the Encapsulation Efficiency

The amount of MTX encapsulated in different formulations was determined by extraction of drug times the extraction efficiency, which were determined by calculating the recovery rate. To calculate the recovery rate, 200 µL of a mixture of polymer: MTX (95:5) in DCM (final concentration of MTX being 1 mg/mL of DCM) was dried under nitrogen flow in four separate tubes. Then, one mL of DCM and 10 mL of distilled water was added to each tube containing dried film and the tubes were closed tightly with Teflon caps. The concentration of MTX in water phase was measured after the tubes were tumbled for one hour at room temperature.

To measure the amount of encapsulated MTX, microspheres were dissolved in DCM at 5 mg/mL in four separate tubes. Then, 9 mL of distilled water was added and tubes capped. After rotating the tubes for one hour, one mL of the top water phase was collected and centrifuged and MTX in supernatant was measured. Then, the encapsulation efficiency (%) was calculated as: (the amount of MTX in the microspheres/the theoretical amount of MTX in the microspheres) × 100.

### 2.6. MTX Release Profile

The release of MTX from PLLA, PLLA + 20% diblock, PLLA + 20% PLGA, and PLGA + 20% MePEG-PDLLA diblock microspheres were obtained at 37 °C in PBS solution (pH 7.4) under sink conditions. To determine the proper formulation with a desirable release profile, ~10 mg of different formulations of MTX containing microspheres were weighed out (n = 4). Then, 10 mL of PBS, pH 7.4 was added to each tube and the tubes were closed tightly, rotated and kept in an incubator at 37 °C, with sample at 1, 2, 4, 24, 48 h and 1, 2, 6, 8, 16, and 22 days followed by the replacement of fresh PBS. The concentration of released MTX was measured using the spectrophotometer as mentioned above. Some early and late release profiles were measured at 1, 2, 3, 5, 8, 15, 21, 27, 30, 33, 39, 45, 50, 60, 66, and 71 days for PLGA + 20%MePEG-PDLLA diblock microspheres.

### 2.7. Evaluation of the Stability of Encapsulated MTX and Released from Microspheres In Vitro

To evaluate the stability of MTX during encapsulation in and release from microspheres, the anti-fibrotic effect of MTX released from loaded microspheres was compared to the freshly made solution of MTX using Western blotting. Supernatant of the polymer-only microspheres soaked in PBS was used as control. Equal amounts of PLGA + 20% MePEG diblock microspheres with or without MTX were incubated in PBS at 37 °C for 24 h. After centrifugation, the concentration of MTX released from microspheres, in supernatant, was measured by spectrophotometry as described above. This solution was then compared to the freshly made solution in these tests.

### 2.8. Fibroblast Cell Culture

Fibroblast culture in DMEM/10%FBS/1% antibiotic, established from neonatal foreskin pieces as previously described [26], at passages 3–7 were used in the experiments conducted in this study (Appendix A). Western blotting for procollagen, MMP-1 or β-actin were performed as described in Appendix A.

### 2.9. In Vivo Animal Studies

To evaluate the anti-fibrotic effect of MTX-loaded microspheres in vivo, a previously established dead space model of wound healing was used [27,28]. Briefly, PVA sponges were hydrated in PBS and sterilized by autoclaving. The dosage of MTX-loaded microspheres reflected MTX efficacy as previously determined [21] and as a function of loaded drug and release characteristics. Then, the required amount of MTX-loaded microspheres and equal amount of control microspheres were weighted and separately loaded in the PVA sponges. Five-week-old male Sprague–Dawley rats weighing 200–250 g (n = 4) were anesthetized (isoflurane) and prepared for implantation. One cm long incisions (Six full thickness, back of each rat) were made and PVA sponges were implanted in a pouch under the panniculus carnosus. There were three groups of implants (2 in each group): (i) PVA sponge implant and no microspheres; (ii) PVA-MTX microspheres; and (iii) PVA-control microspheres. Wounds were sutured and dressed accordingly. Rats were euthanized after 62 days and PVA sponges were collected. PVA sponges with overlying skin from each rat were cut in half and one half was fixed (10% neutral buffered formaldehyde), dehydrated and paraffin-embedded for histology. The overlying skin of the second half was separated and the PVA sponge with tissue grown inside was later used for qPCR following storage at −80 °C. Hydroxyproline was assayed in the second set of whole PVA sponges separated from the overlying skin and stored at −80 °C.

### 2.10. Histological Analysis of Fibrosis

Paraffin-embedded tissues were sectioned for histological evaluation using five micrometer cuts, with rehydration then H and E/Masson’s Trichrome staining [29].

### 2.11. In Vivo Biological Activity; Quantification of Collagen

Quantification of collagen deposited inside and attached to PVA sponges was performed using the hydroxyproline assay as previously described [30] (Appendix A).

### 2.12. Total Tissue Cellularity

Five micrometer-thick sections of paraffin-embedded samples were stained with DAPI for total tissue cellularity. Thirty fields of each PVA sample were imaged using ×200 magnification (Zeiss fluorescent microscope—Axiovision software). Two separate people counted nuclei in 10 fields of coded images from each sample (Image Pro Plus 4.5 software, Media Cybernetics, Inc., Rockville, MD, USA) and the average number of nuclei/fields were used for statistical analysis.

### 2.13. In Vivo Biological Activity; MMP-13, Type-1 Collagen, and α-SMA Gene Expression

The level of type-I collagen, α-SMA expression and MMP-13 in tissue samples was examined by qPCR. Rat MMP-13 is similar to human MMP-1 [31,32]. After RNA extraction (Trizol, Invitrogen), RNA samples (equal concentrations) were reverse transcribed to cDNA (Superscript II First Strand cDNA kit). For qPCR, the SYBR^®^ Green PCR Master-Mix kit was used. Gene expression changes were normalized to PVA control groups.

### 2.14. In Vivo Biological Activity; α-SMA Protein Expression

To examine the α-SMA expression at protein level, previously described methods were used with some modifications [30] (Appendix A). Formalin-fixed, paraffin-embedded, tissue sections were subjected to antigen retrieval and immunohistochemical staining; the only amendment being that the colored product of the enzyme horseradish peroxidase (HRP) was developed with the NovaRED substrate kit for peroxidase followed by hematoxylin counterstaining of nuclei.

### 2.15. Statistical Analysis

In this study, data were expressed as the mean ± SD of three or more independent experiments unless otherwise indicated. To calculate statistical significance, one-way analysis of variance (ANOVA) followed by Scheffe post hoc test was used. The *p* values equal or less than 0.05 were considered statistically significant.

## 3. Results

### 3.1. Morphology and Size Distribution of the Fabricated Microspheres

Scanning electron microscopy images of four different formulations: (i) PLLA 95%; (ii) PLLA 75% + 20% MePEG diblock; (iii) PLLA 75% + 20% PLGA; and (iv) PLGA 75% + 20% MePEG diblock microspheres, each loaded with 5% MTX, are shown in Figure 1A. The PLLA + diblock did not form spherical microspheres (a), PLLA formed small spherical microspheres mixed with particles (b), and only PLLA + PLGA and PLGA + MePEG diblock formed large microspheres (c and d, respectively). However, multiple cracked and broken microspheres were seen when PLLA + PLGA formulation was used for fabrication of microspheres (c).

The measurement of the particle size of the fabricated MTX microspheres using different formulations revealed that PLGA 75% + MePEG- diblock 20% yields larger microspheres with mean particle size of 452.73 ± 47.3 μm, in comparison with PLLA95% (300.7 ± 15.5 μm) and PLLA75% + PLGA20% (331.6 ± 36.1 μm) (Figure 1B). Since PLLA+ diblock formulation did not yield spherical microspheres, the particles were not included in the measurement. The picture shown is Figure 1A(c) illustrate morphology and cracking issues not particle size, which is accurately shown in Figure 1B.

### 3.2. Encapsulation Efficiency and In Vitro Release Profile

Polarized light microscopy images of the four different formulations used to fabricate MTX-loaded microspheres are shown in Figure 2A. The PLLA + MePEG diblock microspheres showed mostly particles with a small amount of MTX inside (yellow green color) (a), the PLLA and PLLA + PLGA microspheres showed more encapsulated MTX (b and c, respectively), and the PLGA + 20% MePEG- diblock showed the most efficient encapsulation. Quantification of the encapsulated MTX in different polymer formulations, as shown in Figure 2B, revealed that the PLLA + MePEG-diblock had the lowest encapsulation efficiency (32.75 ± 6.7%), followed by PLLA with 47.2 ± 1% and PLLA + PLGA with 60.2 ± 5.8%. PLGA + MePEG-diblock with 86.74 ± 8.9% had the highest encapsulation efficiency.

Three formulations which had more than 50% encapsulation efficiency: PLLA, PLLA + PLGA, and PLGA + MePEG-diblock were studied for the early in vitro release profile. As it is shown in Figure 2C(a), PLLA + PLGA released more than 50% of the encapsulated MTX in the first hour (51.4 ± 7.2%) followed by 86.7 ± 7% cumulative release after 48 h. The initial burst release for PLLA was also 35.3 ± 5% for the first hour and reached to 48.7 ± 6.8% at 48 h. PLGA + 20% MePEG- diblock formulation had the lowest initial burst release, 12.4 ± 2% for the first hour and 23.2 ± 4.5% cumulative release after 48 h. While 100% of the encapsulated MTX was released from PLLA + PLGA formulation after 16 days, the cumulative release for PLLA and PLGA + 20% MePEG diblock, after 22 days, were 62 ± 9% and 27 ± 4%, respectively (Figure 2C(b)).

### 3.3. Morphology of the Polymer-Only and Extended-Release Kinetics of MTX-Loaded Microspheres In Vitro

Considering the morphology of the microspheres, the encapsulation efficiency, and the initial release profile, we decided to focus on the PLGA 75% + MePEG-diblock 20% formulation for fabrication of the MTX-loaded microspheres. Therefore, polymer-only microspheres (empty microspheres) with the same formulation were prepared. As shown in Figure 3A, the surface morphology of both empty (a and c) and MTX-loaded microspheres (b and d) appeared to be spherical; however, higher magnification showed that particles of MTX produced small holes on the surface of the MTX-loaded microspheres.

The size distributions of the polymer-only and MTX-loaded microspheres were similar (452.73 ± 47.3 μm). A typical distribution of the MTX-loaded microspheres is shown in Figure 3B to illustrate the excellent low dispersity of the size distribution with very few spheres falling outside the 100 to 1000 μm range. The distribution for non-drug-loaded microspheres was similar (Appendix A).

The extended-release profile for PLGA + 20% MePEG-diblock showed that after the initial burst release there is almost no release for 40 days, followed by a gradual release between days 45 and 50 and a more rapid release between days 50 and 60. Another phase of gradual release happens from day 60 to day 71, when the cumulative released drug reaches 83 ± 7% of the encapsulated one (Figure 3C).

### 3.4. MTX Released from Microspheres Decreases Collagen and Increases MMP-1 Expression at Protein Level in Human Dermal Fibroblasts In Vitro

Equal amounts of PLGA + 20% MePEG-diblock microspheres with or without MTX were incubated in PBS at 37 °C for 24 h. After centrifugation the concentration of the MTX in the supernatant was determined using spectrophotometry. Three different strains of cultured, human dermal fibroblasts were divided into four groups: (i) C: control group without treatment; (ii) EM: treated with equal volume of the supernatant of empty microsphere; (iii) MTX: treated with 50 ng/mL of freshly made MTX solution; and (iv) MTX mic: treated with 50 ng/mL of MTX released from microspheres. The Western blot analysis of the cell lysates showed that MTX released from microspheres has the same effect on the expression of type I procollagen at protein level as freshly made MTX solution (Figure 4A). Quantitative analysis confirmed that MTX released from microspheres significantly decreased the expression of type I procollagen in dermal fibroblasts (17.9 ± 31.1%) in comparison with untreated or treated with the supernatant of the empty microspheres (100%, 106.2 ± 15.8%, respectively), n *=* 3, *p =* 0.004 (Figure 4B).

Additionally, MTX released from microspheres decreased the MMP-1 expression at protein level similar to the freshly made MTX solution (Figure 4C), and quantitative analysis of the relative MMP-1 expression, normalized to β-actin, as loading control, showed that MMP-1 is expressed 2–10 folds more in cells treated with MTX, freshly made or released from microspheres, than untreated or treated with empty microspheres’ supernatant. n *=* 3, *p* = 0.03 (Figure 4D).

### 3.5. MTX Microspheres Decrease Total Cellularity inside Implanted PVA Sponges

Sections of the samples harvested after 62 days and stained with H&E showed comparable growth of new tissue inside the PVA sponges in all three groups (Figure 5A). However, quantification of the total tissue cellularity inside the dead space of the PVA sponges, calculated as mentioned above, revealed that the total number of cells in the PVA alone group (519.5 ± 25.5/field) was significantly higher than the PVA+ empty microspheres (445.3 ± 27.9/field) and PVA + MTX microspheres (400.3 ± 22.6/field) (n = 3, *p* < 0.05). No statistically significant difference was observed between PVA+ empty microspheres and PVA + MTX microspheres (Figure 5B,C).

### 3.6. MTX Microspheres Decrease Collagen Deposition In Vivo

The samples stained with Masson’s trichrome at 62 days after subcutaneous implantation of PVA sponges on the back of the rats revealed that less collagen was deposited inside the PVA sponges that was loaded with MTX-containing microspheres in comparison with the PVA alone or PVA+ empty microspheres (Figure 6A). The relative collagen content, measured by hydroxyproline assay, deposited inside, and attached to the PVA sponges loaded with empty microspheres or MTX microspheres to the PVA sponges alone, was 71.35 ± 23.9% and 56.9 ± 14.5%, respectively (Figure 6B). This decrease was shown to be statistically significant (n = 4, *p* < 0.05).

### 3.7. Col1α1, MMP-13, and α-SMA Expression In Vivo

RNA extraction and qPCR analysis were used to determine the effect of the controlled release of MTX on the expression of Col1α1, MMP-13, and α-SMA in rat samples harvested after 62 days, with changes in gene expression being normalized to PVA control group. The expression of MMP-13 was significantly increased in PVA + MTX microspheres (13.9 ± 8.8- fold) in comparison with the PVA alone or PVA+ empty microspheres (Figure 7A). Interestingly, α-SMA expression was decreased in both PVA+ control microspheres (0.35 ± 0.37) and PVA + MTX microspheres (0.19 ± 0.04) in comparison with the PVA alone group. No statistically significant difference was observed in α-SMA expression between two PVA groups loaded with either empty or MTX containing microspheres. The expression of the Col1α1 was the same among three groups.

The reduction in α-SMA expression in both PVA + MTX and PVA+ empty microsphere groups compared to the PVA alone was further established by immunohistochemical staining (Figure 7B).

## 4. Discussion

Fibrotic scar formation is the final stage of a repair process following tissue injury. While enough collagen deposition as the key component of the fibrotic tissue is required for complete and stable repair of a wound, excessive fibrosis, as seen in hypertrophic scarring or post-surgical adhesions and capsular contractures, is a serious problem with a wide range of physical, emotional, and economical consequences for patients and health care systems [33,34,35,36]. Extensive ongoing research is trying to address the lack of completely satisfactory preventive or therapeutic modality for these fibrotic conditions. Systemic anti-inflammatory agents have been studied in animals with limited success in preventing implant-induced capsule formation [37,38,39]. MTX is a widely used medication that has anti-inflammatory effects when used at low doses [3,40]. In addition, we have previously shown that low dose MTX has an antifibrotic effect, as witnessed by studies in human dermal fibroblasts showing changes in collagen and MMP-1 expression [21]. Unfortunately, serious side effects may be observed with the systemic use of this drug, and topically applied MTX has limited permeability through intact skin or is rapidly cleared from tissue following local injection [12,13]. Therefore, controlled release drug delivery systems may provide the delivery of therapeutic doses of MTX over extended times.

PLLA has been previously used to encapsulate MTX in slow-release microspheres for intra-articular injection [14,41]. For intraarticular injection, a faster degrading polymer like PLLA with a low molecular weight (2K) was required [14]. For other applications PLGA with slower degradation times (2–6 months depending on LA:GA ratio and molecular weight) may be suitable, and PLGA is known to be particularly suitable for manufacturing drug-loaded microspheres [24,25]. The features of the microspheres, such as encapsulation efficiency and drug release patterns, may depend on various parameters involved in the process of fabrication such as physico-chemical characteristics of the encapsulating-polymer, polymer–drug interaction, continuous-phase properties, and continuous phase/dispersed phase ratio [42,43]. In our study, the results of the SEM imaging of the MTX-loaded microspheres fabricated by using different polymer/blends showed that while low molecular weight PLLA (2K) can form microspheres, the addition of MePEG-diblock to PLLA 2K yields only particles without spherical microsphere formation. The combination of PLLA 2K and PLGA or PLGA and MePEG-diblock yielded larger microspheres, under the same experimental condition. It has been suggested that for a given stirring rate the organic phase droplets may be subject to shear stress and break up, and this may be influenced by the viscosity differences in the use of organic solvents [14]. Since the intrinsic viscosity of the PLLA and PLGA polymers used in this study were 0.1–0.2 dL/g and 0.61 dL/g, respectively, the higher viscosity of the organic phase solutions containing PLGA may have resulted in larger microspheres due to higher resistance to droplet breakoff. Additionally, our results showed that although acceptable encapsulation of MTX in PLLA 95%, PLLA 75% + PLGA 20%, and PLGA 75% + MePEG-diblock 20% was achieved, the combination of PLGA 75% + MePEG-diblock 20% yielded microspheres with the highest drug encapsulation efficiency. The solidification rate of polymer droplets in the emulsion is one of the important factors affecting the entrapment of drugs in different polymers [42]. In a previous study, MTX was encapsulated in microspheres fabricated from PLLA 2K, 50K, and 100K [14]. The authors observed that the precipitation and hardening rate of the PLLA 50K and 100K as microspheres was faster than the PLLA 2K, allowing for the retention of the MTX and higher encapsulation efficiency [14]. In our study, the high molecular weight (MW) of the PLGA 85/15 (90K–126K) likely resulted in faster solidification of the microspheres and higher encapsulation efficiency. Moreover, the addition of the MePEG-diblock to PLLA resulted in reduced encapsulation efficiency as expected from previous studies [24].

Along with the encapsulation efficiency, the profile of release of the encapsulated agent is an important factor that determines successful clinical use of such formulations for delivering drugs. The drug release from PLLA and PLGA polymers is biphasic: starting with fast release from surface deposits of drug, and a second phase during which a progressive drug release is seen because of both diffusion and polymer degradation [14,24,44]. Factors affecting the rate of initial burst release are related to drug type, drug concentration, and hydrophobicity of the polymer [24]. Other factors such as preparation techniques, porosity of the microspheres, and polymer blending methods can also impact release characteristics [44]. Since excessive fibrotic tissue formation happens in the later stages of healing, MTX-loaded microspheres with less fast starting release and a secondary slower and delayed release seem more suitable for prevention of post-surgical or implant-induced fibrosis. In this study it has been demonstrated that the microspheres fabricated with PLGA 75% + MePEG-diblock 20% have the lowest initial burst release in comparison with PLLA and PLLA + PLGA formulations. The higher initial burst release of microspheres fabricated with PLLA 75% + PLGA 20% may be explained partly because of the presence of cracked and broken microspheres evident in SEM images. The low solubility of MTX in DCM and dispersion of particulate MTX in polymer/DCM solution leads to entrapment of MTX as particles in polymer matrix. Differential scanning calorimetry has previously reinforced the possible solid dispersion of MTX in PLLA microspheres and suggested as a contributing factor to rapid-starting drug release [14]. Cracked and broken microspheres may increase the exposed surface area to buffer resulting in higher burst release. The results of the extended kinetics of PLGA + MePEG-diblock microspheres with entrapped MTX showed a relatively long period of negligible release of MTX following initial burst release and a second phase of more constant release in accordance with previous studies [24].

One advantage generally of using blends of PLGA with diblock copolymers is that for very hydrophobic drugs the diblock polymers may micellize drugs and modulate drug release [23,45,46]. However, a more important advantageous aspect for the use of diblocks in polymeric matrices used in potentially inflammatory tissue areas (such as wounds) might be that diblock-containing PLGA microspheres are less inflammatory than PLGA-alone microspheres [23]. This effect arises from the preferred orientation of the diblock copolymers at the microsphere surface reducing hydrophilicity, opsonization and immune cell activation [23].

High tissue cellularity is one of the histological characteristics of fibrotic conditions. In this study we found that MTX-loaded microspheres reduced total tissue cellularity in comparison to the PVA control group, which may be possibly due to the anti-inflammatory effect of MTX [40]. It is also possible that lactic acid accumulation from water-based degradation of PLGA may be a factor that explains the reduced total number of cells in PVA+ control microspheres group compared to the PVA alone; this is in agreement with a previous study showing that lactic acid may inhibit G1/S keratinocyte cell cycle progression with induction of apoptosis [47].

Fibrotic conditions are characterized by excessive collagen deposition which results from abnormalities in collagen production and degradation by MMP-1. In this study, dermal fibroblasts that were treated with MTX (released from PLGA+ diblock microspheres) resulted in decreased collagen and increased MMP-1 production, similar to the freshly made MTX solution. Furthermore, the embedded MTX-loaded microspheres significantly reduced the collagen deposition in implanted PVA and increased the MMP-13 expression (equivalent to MMP-1 in humans) compared to the PVA control and PVA+ control microspheres. Previously reported MTX induced reduction in type-I collagen and increases in MMP-1 expression in primary dermal fibroblasts are in sync with the findings of this study where the release of MTX is delayed and slow from microspheres [21].

In addition to modulation of collagen and MMP-13 expression, MTX-loaded microspheres reduced the α-SMA expression, both at gene and protein level, reflected in the reduced number of α-SMA expressing cells, when compared to the PVA alone group. While reduction in α-SMA mRNA expression in PVA + MTX microspheres might be partly due to the anti-fibrotic effect of MTX, locally produced lactic acid might also contribute, which similarly could explain the significantly lower α-SMA mRNA expression and protein in PVA+ control microspheres in comparison with PVA alone group.

## 5. Conclusions

Taken together, our results show that MTX can be efficiently encapsulated and released from PLGA + MePEG-diblock microspheres. Implantation of MTX-loaded microspheres reduced fibrosis in vivo. The fabricated MTX microspheres have the potential to be used not only for prevention or treatment of fibrotic conditions, but also for other inflammatory diseases. Further investigation is needed to elucidate the possible role of lactic acid, produced from biodegradation of PLGA polymer microspheres, on tissue cellularity and α-SMA expression.

The limitation of the in vivo part of this investigation was the low numbers of animals included, and so it should be considered as a pilot study. Further studies with larger numbers and including other animal models of fibrosis are suggested.

## Figures and Tables

**Figure 1 bioengineering-10-00298-f001:**
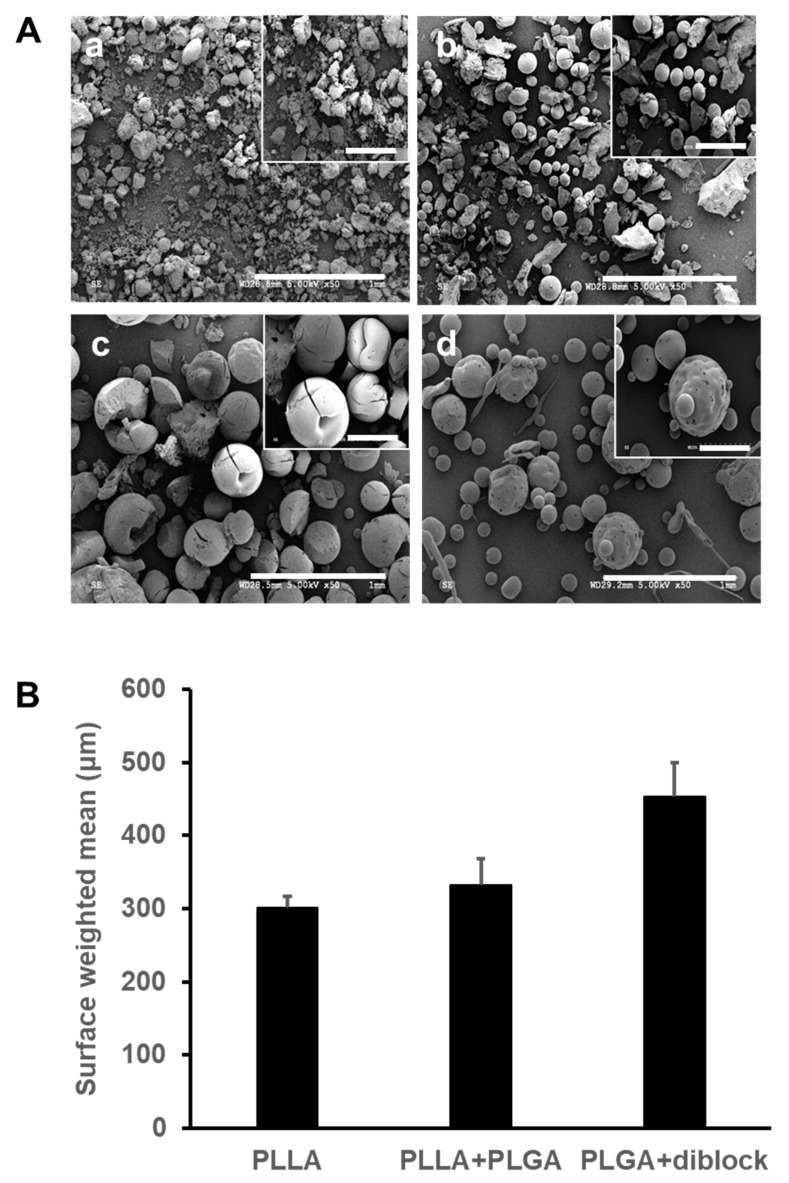
Scanning electron microscopy images and size of four different polymer formulations loaded with MTX. (**A**) PLLA 75% + MePEG diblock 20% formed particles (**a**), PLLA 95% formed small spherical microspheres mixed with particles (**b**), PLLA 75% + PLGA 20% (**c**), and PLGA 75% + MePEG-diblock 20% (**d**) formed mostly spherical microspheres. Multiple cracked and broken microspheres were seen with PLLA + PLGA formulation (**c**). Scale bar = 1 mm in main image and 300 μm in insert. (**B**) surface weighted mean is reported as the mean diameters. The values are shown as the mean of four batches of microspheres ± one standard deviation.

**Figure 2 bioengineering-10-00298-f002:**
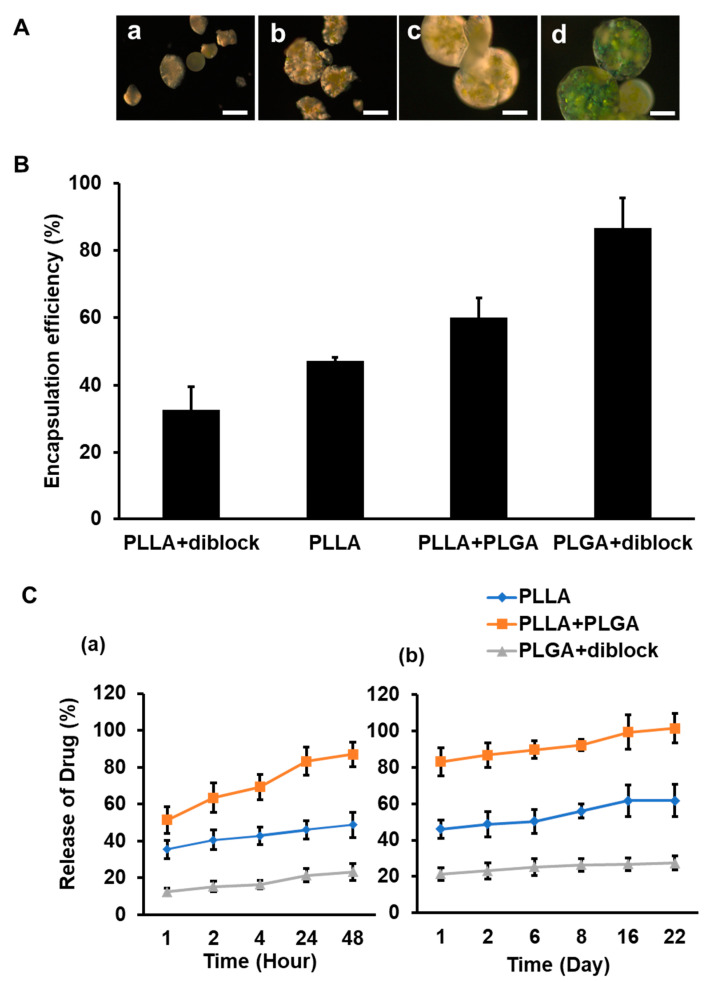
Encapsulation efficiency and in vitro drug release profile from the microspheres. (**A**) Polarized light microscopy images of MTX encapsulated in PLLA + MePEG-diblock (**a**), PLLA (**b**), PLLA + PLGA (**c**), and PLGA + MePEG- diblock polymer microspheres (**d**), scale bar = 200 μm. (**B**) Quantification of the encapsulation efficiency of four separate batches of each formulation revealed that PLLA + MePEG- diblock had the least and PLGA+ diblock had the highest encapsulation efficiency. (**C**) Comparison of three different formulations of the polymer showed that PLLA + PLGA microspheres had the highest initial burst release at the first 48 h (**a**), and almost 100% of the encapsulated MTX was released from them after 16 days (**b**). PLGA + MePEG-diblock microspheres had the lowest initial burst release both at the first 48 h (**a**) and after 22 days (**b**).

**Figure 3 bioengineering-10-00298-f003:**
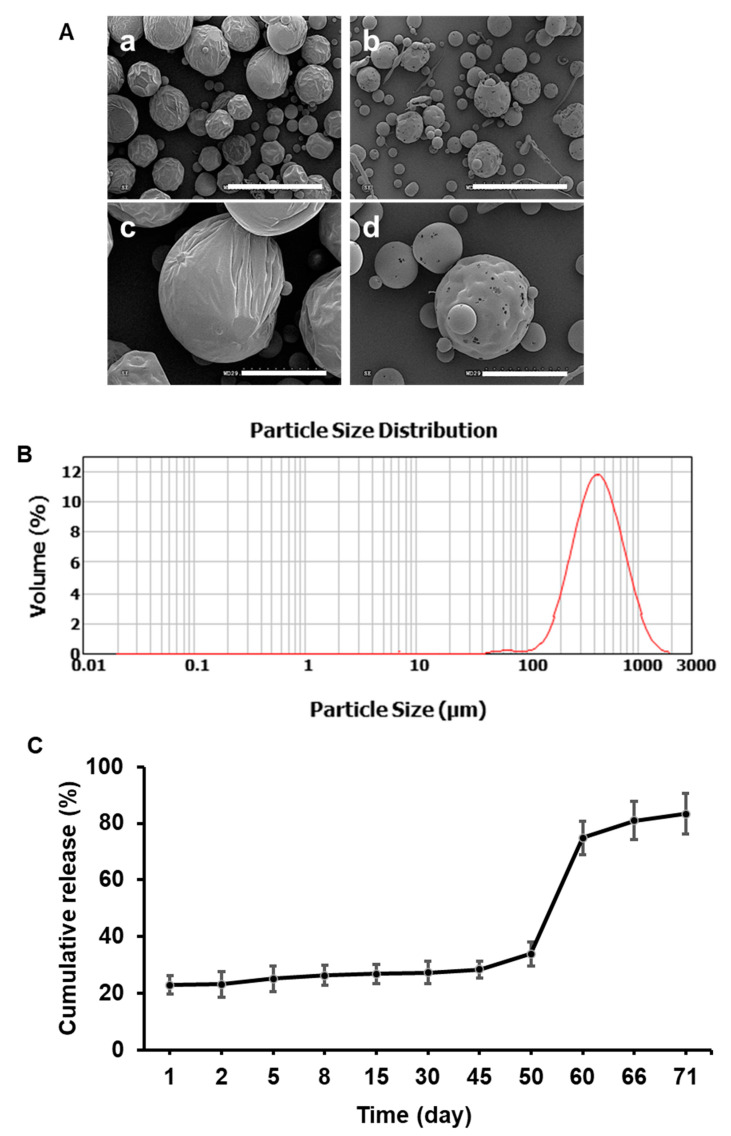
Morphology of the polymer-only and MTX-loaded PLGA + MePEG-diblock microspheres and extended-release profile in vitro. (**A**) Empty (**a**,**c**) and MTX-loaded PLGA 75% + MePEG-diblock (**b**,**d**) microspheres are spherical. High magnification shows that MTX-loaded microspheres have small holes on the surface. Scale bar = 1mm in (**a**,**b**) and 300 μm in (**c**,**d**). (**B**) Representative graph of particle size distribution of the MTX (5%) loaded PLGA 75% + 20% MePEG-diblock microspheres. (**C**) Extended-release profile of PLGA + MePEG-diblock microspheres showed low initial burst release, followed by a gradual release after 40 days. A second burst release happened over 10 days between days 50 and 60.

**Figure 4 bioengineering-10-00298-f004:**
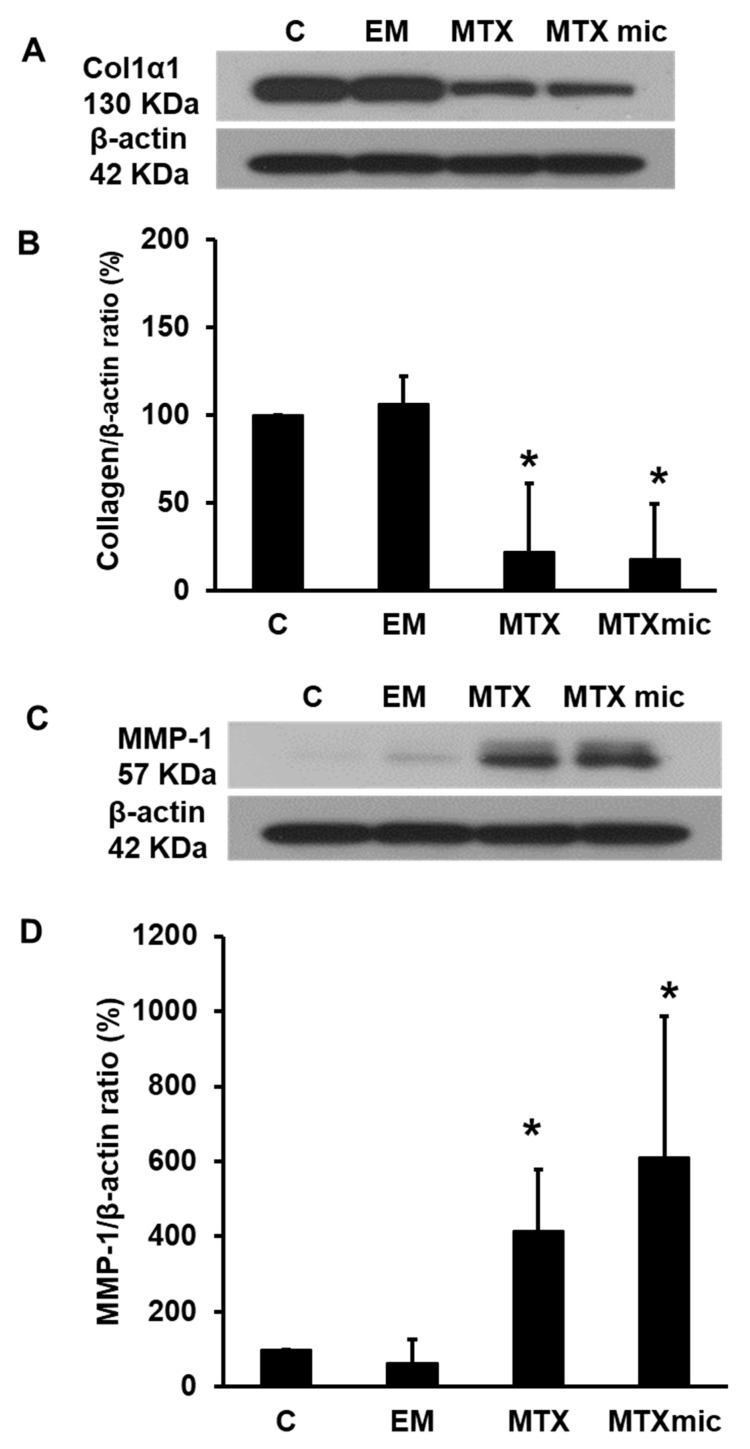
The effect of MTX released from microspheres on type I procollagen and MMP-1 expression in primary human dermal fibroblasts (**A**,**C**). Representative Western blot of cell lysates of fibroblasts treated with 50 ng/mL of freshly made MTX solution or MTX released from microspheres. Quantitative analysis of the relative procollagen (**B**) and MMP-1 (**D**) expression, normalized to β-actin as loading control, revealed that cells treated with MTX (freshly made or released from microspheres) express significantly less type I procollagen and more MMP-1 than untreated or treated with supernatant of empty microspheres. Data represent the mean ± SD for n = 3. * Statistical significance, *p =* 0.004.

**Figure 5 bioengineering-10-00298-f005:**
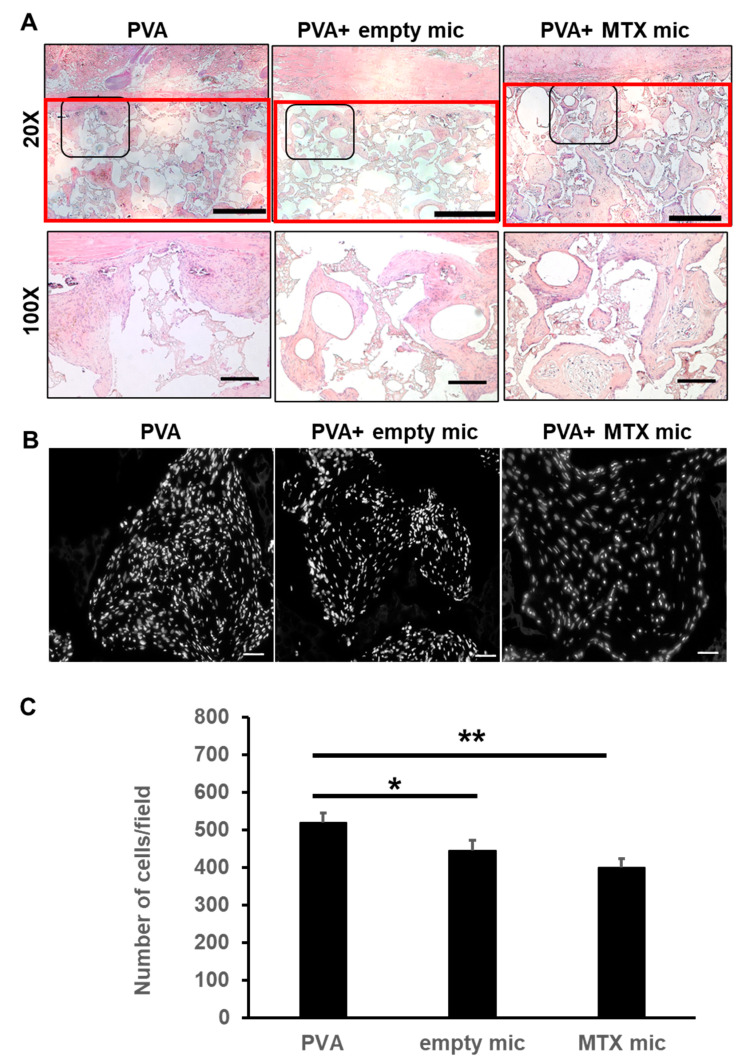
Tissue cellularity and growth in PVA sponges from skin samples after 62 days. (**A**) H&E stained sections show comparable new tissue growth red marking area) for all three groups: (i) control PVA; (ii) PVA containing control microspheres; and (iii) PVA + MTX microspheres. Scale bar = 1 mm at 20 and 200 μm at 100 magnifications. (**B**) Representative images of the sections (DAPI stained, 200×, 50 μm scale bar). (**C**) Quantitative analysis of the number of cells in the interior of PVA showed that the total number of cells/fields in PVA control images (519.5 ± 25.5 cells/field) was significantly higher than the PVA+ control microspheres (445.3 ± 25.5 cells/field) and PVA + MTX microspheres (400.3 ± 22.6 cells/field). n = 3, * *p* = 0.05, ** *p* = 0.0007.

**Figure 6 bioengineering-10-00298-f006:**
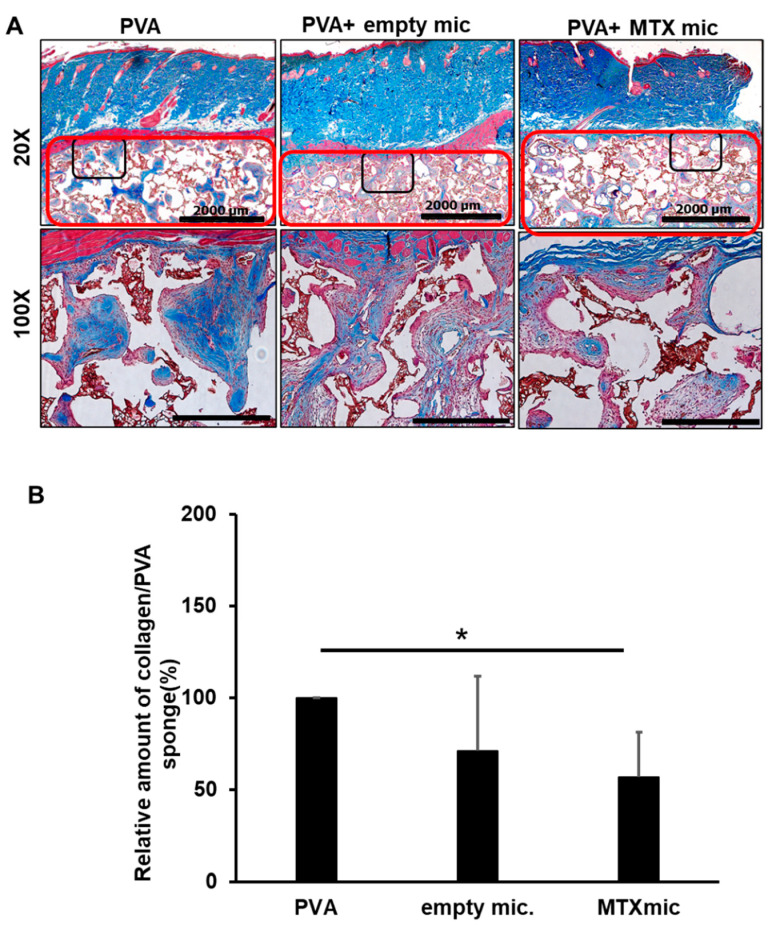
Collagen deposition in the PVA sponges after 62 days. (**A**) There was less collagen deposition in PVA + MTX microspheres compared to PVA alone and PVA+ empty microspheres (Reflected as less blue color in areas marked by red drawing, Masson’s trichrome staining). Scale bar = 2 mm at 20 and 500 μm at 100× magnification. (**B**) Quantification of the collagen showed that MTX containing microspheres significantly decreased collagen deposition in PVA + MTX microspheres in comparison with PVA alone samples. n = 3, * Statistical significance, *p <* 0.05.

**Figure 7 bioengineering-10-00298-f007:**
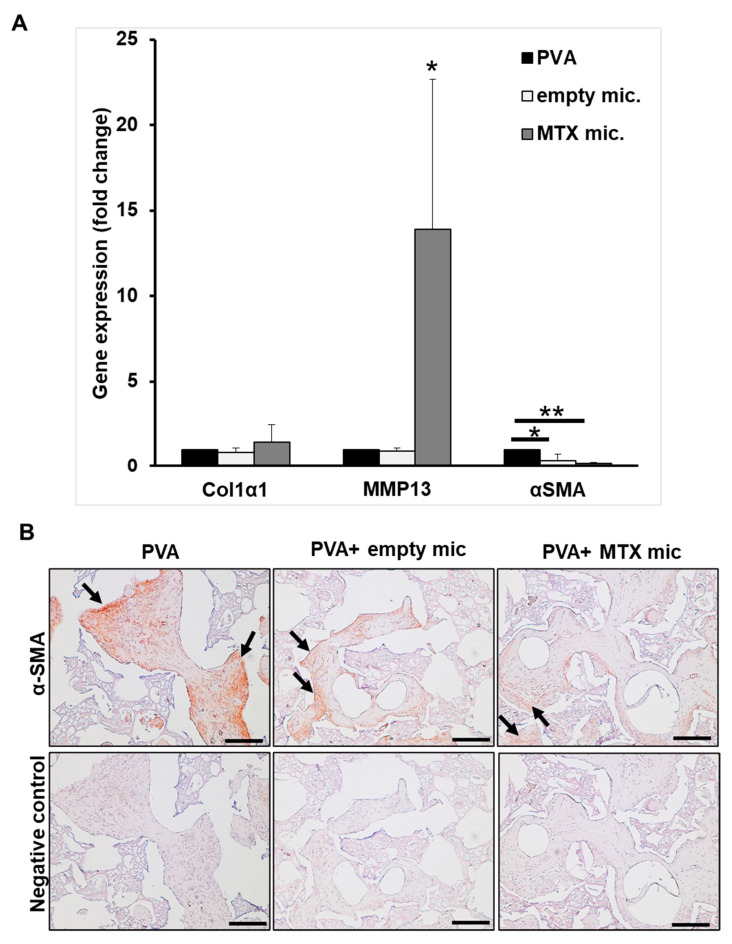
Col1α1, MMP-13, and α-SMA expression in rats after 62 days. (**A**) qPCR results revealed that MTX containing microspheres significantly increased the MMP-13 expression (13.9 ± 8.8-fold) in comparison with the PVA alone or empty microspheres. The expression of α-SMA decreased significantly in both empty and MTX microsphere-loaded PVAs compared to the PVA alone group. n = 3, *** indicates statistical significance, * *p* < 0.05, ** *p* = 0.004. (**B**) Immunohistochemical (IHC) staining of samples revealed decreased α-SMA expression (Arrows), in PVA + MTX and PVA+ empty microspheres compared to the PVA alone group. Scale bar = 200 μm.

## Data Availability

Not applicable.

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
