# Peer review of "Novel, Blended Polymeric Microspheres for the Controlled Release of Methotrexate: Characterization and In Vivo Antifibrotic Studies"

_bioengineering, 2023, doi:10.3390/bioengineering10030298_

Round 1

Reviewer 1 Report

The manuscript titled “Novel, blended polymeric microspheres for the controlled release of methotrexate: Characterization and in vivo antifibrotic studies” describes the fabrication of MTX-loaded polymeric microspheres for the prevention of fibrosis in vivo. The manuscript is well-written and provides interesting findings.

I only have the following remarks to be considered:

1-A graphical representation of the microspheres compositions will be very interesting and informative for the readers.

2-In the Experimental section: please, describe the experimental methods and procedures in more detail, and do not refer to a previous method e.g. Histological Analysis of Fibrosis, In Vivo Biological Activity; Quantification of Collagen, In Vivo Biological Activity; MMP-13, Type-1 Collagen, and α-SMA Gene Expression, In Vivo Biological Activity; α-SMA Protein Expression, and so on. Other researchers in the field should be able to reproduce your findings. You can even describe your methods in a supplementary file.

Reviewer 2 Report

1.      The manuscript is hard to read. It is necessary to rewrite it.

2.      The abbreviation of formulation must be consistent. Ex: PLLA+ Me-PEG deblock, PLLA+ 20% diblock, PLLA 75%+ 20% MePEG diblock………

3.      Line 87: PLGA (85/15, IV=0.61 dL/g)

What is IV?

4.      Line 251-252 “PLGA 75% + MePEG- diblock 20% yields larger microspheres with mean particle size of 452.73± 47.3 μm in comparison with PLLA95% (300.7±15.5 μm) and PLLA75%+PLGA20% (331.6±36.1 μm) (Figure 1B).”

However, PLLA75%+PLGA20% showed a larger diameter than PLGA 75% + MePEG- diblock 20% in Figure 1B.

5.      There are many typos in the manuscript. Ex: Line 271: 32.75%± 6.7 need correct to 32.75± 6.7%.

6.      It is lack of scale bar in Figure 2A

7.      It is necessary to add the particle size distribution of the PLGA75% + 20% MePEG diblock microspheres in figure 3B to demonstrate “The size distributions of the polymer only and MTX loaded microspheres were similar (452.73± 47.3 μm) (Figure 3B).”

8.      80% reference papers are older than 5 years old.

Reviewer 3 Report

This manuscript develops blended polymeric microspheres for the controlled release of methotrexate. The manuscript was clearly written, and the data were properly presented. Some minor concerns still need to be addressed before it is suitable for publication.

Minor concerns:

1. In the efficacy study, the individual variation for in vivo experimental design including number of animals per group (n =4) below 6 (even 5) was unclear and untenable, the justification for statistical analysis was may not validated. The assay needs more rats to demonstrate the efficacy.

2. What’s the cellular uptake mechanism of the nanoparticles? A flow cytometry or LC-MS/MS assay need to construe the mechanism, see for example the below published article:

Camptothesome elicits immunogenic cell death to boost colorectal cancer immune checkpoint blockade. Journal of Controlled Release 349 (2022) 929-939

3. The current manuscript only demonstrated the anti-fibrotic formation effect in vivo, how about the efficacy in a preform fibrotic animal model.

Author Response

please see attached response

Round 2

Reviewer 1 Report

The author addressed my comments, and the paper is eligible to be accepted.

Reviewer 2 Report

The authors have addressed my comments. it is suitable to publish in bioengineering